# Asynchronous Embryo Transfer Followed by Comparative Transcriptomic Analysis of Conceptus Membranes and Endometrium Identifies Processes Important to the Establishment of Equine Pregnancy

**DOI:** 10.3390/ijms21072562

**Published:** 2020-04-07

**Authors:** Charlotte Gibson, Marta de Ruijter-Villani, Stefan Bauersachs, Tom A.E. Stout

**Affiliations:** 1Department of Clinical Sciences, Faculty of Veterinary Medicine, Utrecht University, 3584CM Utrecht, The Netherlands; chrlt.gibson@gmail.com (C.G.); M.Villani@uu.nl (M.d.R.-V.); 2Institute of Veterinary Anatomy, Vetsuisse Faculty Zurich, University of Zurich, 8315 Lindau (ZH), Switzerland; stefan.bauersachs@uzh.ch

**Keywords:** horse, asynchronous embryo transfer, conceptus, endometrium, transcriptome, exosome

## Abstract

Preimplantation horse conceptuses require nutrients and signals from histotroph, the composition of which is regulated by luteal progesterone and conceptus-secreted factors. To distinguish progesterone and conceptus effects we shortened the period of endometrial progesterone-priming by asynchronous embryo transfer. Day 8 embryos were transferred to synchronous (day 8) or asynchronous (day 3) recipients, and RNA sequencing was performed on endometrium and conceptuses recovered 6 and 11 days later (embryo days 14 and 19). Asynchrony resulted in many more differentially expressed genes (DEGs) in conceptus membranes (3473) than endometrium (715). Gene ontology analysis identified upregulation in biological processes related to organogenesis and preventing apoptosis in synchronous conceptuses on day 14, and in cell adhesion and migration on day 19. Asynchrony also resulted in large numbers of DEGs related to ‘extracellular exosome’. In endometrium, genes involved in immunity, the inflammatory response, and apoptosis regulation were upregulated during synchronous pregnancy and, again, many genes related to extracellular exosome were differentially expressed. Interestingly, only 14 genes were differentially expressed in endometrium recovered 6 days after synchronous versus 11 days after asynchronous transfer (day 14 recipient in both). Among these, *KNG1* and *IGFBP3* were consistently upregulated in synchronous endometrium. Furthermore bradykinin, an active peptide cleaved from KNG1, stimulated prostaglandin release by cultured trophectoderm cells. The horse conceptus thus responds to a negatively asynchronous uterus by extensively adjusting its transcriptome, whereas the endometrial transcriptome is modified only subtly by a more advanced conceptus.

## 1. Introduction

Early pregnancy in the horse is characterized by an unusually long preimplantation period, encompassing a number of enigmatic events [1]. In particular, the molecule or stimulus that the horse conceptus uses to signal its presence and ensure maternal recognition of pregnancy (MRP) has not yet been identified, nor do we fully understand how the conceptus guides endometrial preparation for implantation. Soon after arriving in the uterus on day 6 after ovulation [2], the equine embryo becomes completely enveloped by an acellular glycoprotein capsule, which has been proposed to help it remain unattached and mobile within the uterine cavity and to act as a ‘mail box’ for regulating the exchange of signaling molecules [3,4,5]. Conceptus mobility persists until day 16 of gestation [6] and appears to be essential for MRP, since restricting migration to a single uterine horn results in failure to avert luteolysis [7]. In the absence of pregnancy, luteolysis occurs between approximately days 13 and 16 as a result of pulses of prostaglandin F2α (PGF2α) released by the endometrium following the establishment of an oxytocin-PGF2α positive feedback loop [8,9]. To prevent luteal regression, the embryo’s MRP signal must prevent the cyclical increase in oxytocin sensitivity that starts at approximately day 10 after ovulation [10,11]. Although the identity of the equine embryonic MRP signal(s) remains unknown [12], it is increasingly clear that MRP involves the transient suppression of PGF2α release by a combination of inhibiting prostaglandin-endoperoxidase-synthase 2 expression (PTGS2; the rate-limiting enzyme in PGF2α synthesis) and delaying the upregulation of the endometrial oxytocin receptors critical to the oxytocin-PGF2α loop needed to generate luteolytic pulses of PGF2α [8,10,13].

Recent transcriptomic and proteomic analyses have improved our understanding of embryo-maternal communication during the conceptus migration and MRP phase. Studies have compared gene expression in endometrium between cycling and pregnant mares on days 8 and 12 [14,15], days 9, 11 and 13 [16], days 12, 14, 16 and 18 [17] or on day 13.5 [18], and between conceptus membranes on days 8, 10, 12 and 14 [19]. A further study looked for possible relationships (e.g., ligand–receptor associations) between the endometrial and conceptus transcriptomes on day 16 [20]. The proteomic profile of day 9 and 10 blastocyst secretions and blastocoele fluid [21], and day 13 uterine luminal (ULF; cyclic and pregnant mares) and yolk sac (blastocoele) fluid have also been reported [22]. The latter studies identified differences in the ULF induced by the presence of a conceptus, and identified proteins secreted by the endometrium or the conceptus into the ULF or yolk sac fluid. Gene ontology analysis for the endometrial transcriptome has revealed that early pregnancy is associated with changes in biological processes including “angiogenesis/vascular remodeling”, genes influenced by steroid hormones (estradiol and progesterone) and by prostaglandin signaling [15], and genes related to cell communication, cell adhesion and response to stimuli [20]. In the conceptus, genes involved in developmental processes and in responses to external stimuli (hormones and endometrial factors) were highly represented [20]. Other processes, such as “lipid biosynthetic processes” and “fatty acid metabolic process” were also upregulated during conceptus development [19]. Comparative proteomics similarly highlighted changes in functional categories of ULF proteins including developmental processes, “binding”, response to a stimulus and lipid and cholesterol activity [21,22].

Besides the conceptus, the major regulators of endometrial physiology and the establishment of uterine receptivity for implantation are the ovarian steroid hormones estrogens and, in particular, progesterone (P4) [23]; P4 helps maintain or alter the uterine environment in ways favorable to embryo growth and development, e.g., by regulating histotroph quality and quantity [24,25]. In sheep and cattle, induction of an earlier than normal post-ovulatory rise in progesterone (by exogenous P4 administration) leads to an earlier onset and more rapid elongation of the conceptus and secretion of the MRP factor (interferon τ); early post-ovulation P4 administration has therefore been used to try to distinguish the effects of P4-priming on endometrial function and the development of receptivity for implantation from those induced by embryo-secreted factors [26,27,28]. In horses, the postovulatory rise in P4 is much more rapid (approximately 24 h after ovulation [29]) than in cattle (approximately 5 days [30]) such that there is very little scope for advancing the onset of the progesterone rise to accelerate conceptus development or identify P4-induced endometrial factors that promote conceptus growth. Moreover, simply raising the P4 concentration during early diestrus by providing exogenous P4 does not appear to affect endometrial gene expression or function [31,32]. An alternative approach to temporally separating P4 and conceptus-induced effects on endometrial function is asynchronous embryo transfer (ET), i.e., transferring an embryo to a more advanced (positive asynchrony) or retarded (negative asynchrony) uterine environment. In sheep and cattle, an embryo-endometrium asynchrony of as little as 2 days either retards (negative asynchrony) or accelerates (positive asynchrony) conceptus development; however, both compromise embryo viability and increase the incidence of pregnancy loss [33,34,35]. By contrast, horse embryos tolerate a remarkable degree of negative (≥5 days) but not positive (≤2 days) uterine asynchrony [36]. Indeed, in clinical ET programs, recipient mares ovulating between 1 day before and up to 5 days after the donor offer similar likelihoods of establishing and maintaining pregnancy [37,38]. However, while horse embryos can be transferred to recipient mares that ovulated up to 5 days after the donor without compromising embryo survival, conceptus development is markedly delayed by asynchrony of more than 4 days [36,39,40]. For example, the embryonic disc (day 14) and embryo proper (day 19) are smaller and less well developed in conceptuses resulting after transfer of day 8 horse embryos into a 5-day negatively asynchronous recipient compared to a synchronous recipient [40]. In addition, the pregnancy stage-related upregulation in mRNA expression for a range of imprinted genes, glucose and amino acid transporters is delayed in conceptuses that develop in a negatively asynchronous uterus [40,41,42].

In the current study, day 8 embryos were transferred to recipient mares that ovulated either on the same day (synchronous) or 5 days after (asynchronous) the donor, and RNA sequencing (RNA-seq) was performed to compare the transcriptome in endometrium and conceptus membranes recovered 6 and 11 days after ET. In this way, the endometrium of asynchronous recipients would be exposed to a shorter period of P4 priming, but an identical period of conceptus interaction; the hypothesis was that the difference in the duration of P4 priming would enable us to dissect some of the roles of progesterone and the embryo in regulating uterine preparation for implantation and signaling to the conceptus. Two of the molecules identified by the RNA-seq analysis as consistently upregulated in the endometrium of synchronous compared to asynchronous recipients, namely kininogen 1 (*KNG1*) and insulin-like growth factor binding protein 3 (*IGFBP3*), were selected for functional studies. We hypothesized that these two molecules would stimulate conceptus prostaglandin secretion (KNG1), cell adhesion and proliferation (IGFBP3) and thereby contribute to the establishment of uterine receptivity and/or help the conceptus match its developmental stage to that of the uterus.

## 2. Results

### 2.1. RNA-seq Analysis and Gene Ontology

After initial analysis of endometrial transcript data, pair-wise correlation heat maps were used to cluster the data from individual samples; this was followed by hierarchical clustering (HCL) of the log2 fold changes of differentially expressed genes (DEGs). From the HCL, different general patterns of expression could be observed between sample groups (Figure 1). Endometrial samples 9, 10, 11 and 12 (day 19 synchronous) clustered together, as did samples 5, 6 and 7 (day 14 asynchronous). On the other hand, samples 1, 2 and 4 (day 14 synchronous) and samples 13, 14 and 15 (day 19 asynchronous) were intermingled in one large cluster with similar expression profiles; the common denominator for these samples was that they were all recovered from mares on day 14 after ovulation. Two samples, 3 and 8, exhibited a very different pattern of expression to the other members of their respective groups (i.e., were outliers) and were therefore not included in subsequent statistical analyses. Looking back through the clinical data, these two samples were derived from day 8 embryos recovered from the same donor mare at different cycles, but were comparable in size to all other day 8 embryos and gave rise to pregnancy vesicles that did not differ in size to other group members. Both recipient mares had two corpora lutea; however, so did at least one other mare per group, i.e., there were no clinical data to explain why these samples were outliers.

We compared endometrial gene expression between synchronous and asynchronous recipients on day 14 (6 days after ET: D14-syn/asyn) and day 19 (11 days after ET: D19-syn/asyn) of conceptus development, and between day 14 synchronous and day 19 asynchronous (D14-sync/D19-asyn; both sets of recipients on day 14 after ovulation). In total, 15,502 transcripts were detected in endometrial samples; these included 523 DEGs for D14-syn/asyn and 715 DEGs for D19-syn/asyn but only 14 DEGs for D19-asyn/D14-syn (fold change >1.4; FDR >5%; P-adjusted <0.05: Appendix A). Among these DEGs 413, 600 and 6 were specific to their respective comparison. The numbers of up- and downregulated genes are shown in Table 1, and the overlap of DEGs between D14-syn/asyn, D19-syn/asyn and D19-asyn/D14-sync is depicted in Figure 2. Only three annotated genes (*IGFBP3*, *KNG1* and *FAM118B*) were differentially expressed between all three comparison groups, while there were five additional DEGs in common between D19-syn/asyn and D14-syn/D19-asyn and 107 DEGs in common between D14-syn/asyn and D19-syn/asyn.

A similar analysis was performed for conceptus membrane transcriptomes. In this case, conceptuses from the same group (synchronous or asynchronous and day of recovery) clustered together with grossly similar patterns of expression, as shown by the pair-wise correlation heat map (Figure 3A) and the HCL (Figure 3B). Day 19 synchronous and day 19 asynchronous conceptuses clustered into two very distinct groups, whereas day 14 synchronous and day 14 asynchronous conceptuses had more similar expression profiles. A total of 13,591 transcripts were detected, of which 556 were classified as DEGs for D14-syn/asyn (fold change >1.4; FDR >5%; P-adjusted <0.05), 3472 DEGs for D19-syn/asyn and 3646 DEGs for D14-sync/D19-asyn with 236, 2239 and 2434 exclusive to the respective pairwise comparisons (Appendix A). The numbers of up- and downregulated genes are shown in Table 1. We identified 98 genes that were differentially expressed in all three comparisons, 121 additional DEGs shared between D14-syn/asyn and D19-syn/asyn, 100 additional DEGs in common between D14-syn/asyn and D14-sync/D19-asyn and 1013 DEGs in common between D19-syn/asyn and D14-sync/D19-asyn (Figure 4).

### 2.2. Gene Ontology Analysis

The DEGs were analyzed using the DAVID Functional Annotation Chart tool for gene ontology (GO) classification, to identify overrepresented functional terms among upregulated and/or downregulated genes in synchronous pregnancy. In the endometrium, the biological processes ‘activation of MAPK activity’, and the cellular components ‘blood microparticle’ and ‘integral component of plasma membrane’ were overrepresented among upregulated genes on both day 14 and 19 of conceptus development (Figure 5). The molecular functions NADP binding and oxidoreductase, and the cellular components lamellipodium were overrepresented among downregulated genes in synchronous pregnancies on both days 14 and 19. Functional terms including proteinaceous extracellular matrix, calcium ion and peptide binding, cell–cell signaling, cell migration and regulation of cell growth were enriched among the genes upregulated in endometrium on day 14 of conceptus development. Functional terms specifically enriched among upregulated genes on day 19 included extracellular exosome, the inflammatory response, immunity, cellular response to vascular endothelial growth, negative regulation of apoptotic process and membrane. Functional terms overrepresented among downregulated genes on day 14 included cellular components such as ion transport and extracellular exosome. On day 19 of conceptus development, in utero embryonic development, cell adhesion, lipid metabolism, oxidoreductase and zinc ion binding were overrepresented among the downregulated genes in synchronous endometrium (Figure 5). Too few genes were differentially expressed between endometrium groups in the D14-sync/D19-asyn comparison to identify functional categories.

In the conceptus membranes, many more functional categories were identified, largely because many more genes were classified as differentially expressed between the synchronous and asynchronous conditions (Figure 6). The categories overrepresented among upregulated genes on both days were extracellular exosome and blood microparticle. Functional categories overrepresented among the upregulated genes in day 14 synchronous conceptuses included the central nervous system and metanephros development, retina layer formation, blood coagulation, regulation of vasoconstriction and regulation of exocytosis and protein secretion. On day 19, categories enriched for upregulated genes included cell adhesion and cell–matrix adhesion, focal adhesion, regulation of cell migration and cell shape and organelles (Golgi apparatus and endoplasmic reticulum). Molecular functions such as calcium ion binding, transferase and kinases, were also enriched in synchronous conceptuses on day 19 of development. In day 14 conceptus membranes, the functional terms overrepresented among downregulated genes of synchronous conceptuses included the biological processes translation and protein heterotetramerization, and other processes related to nuclear function such as nucleosome assembly, DNA binding and the cellular components nucleus, nucleosome core and chromosome. Clusters of downregulated genes on day of 19 of synchronous conceptus development included cellular components related to extracellular exosome, and mitochondria (e.g., mitochondrion, mitochondrial inner membrane and respiratory chain complex I), and components related to the lysosomal membrane and oxidoreductase function and certain metabolic processes. Genes associated with the biological processes ‘defense response to viruses’ and antiviral defense were also overrepresented among the downregulated genes on day 19.

### 2.3. RNA-seq Validation by qRT-PCR

To validate the RNA-seq results, quantitative RT-PCR was performed for 11 genes selected specifically for each tissue (Table 2: endometrium; Table 3: conceptus). We compared gene expression on day 14 and 19 of conceptus development between synchronous and asynchronous ET, and measured the log2 fold change. Overall, changes in gene expression (up- or downregulated) identified by RNA-seq (P-adjusted < 0.05) were confirmed by qRT-PCR, although the difference was not always statistically significant in the qRT-PCR comparisons (*p*-value > 0.05).

In endometrial samples, *LIFR*, *SLC43A2* and *SLC5A1* transcripts showed no significant difference between conditions in the RNA-seq analysis, findings that were replicated by qRT-PCR. In synchronous endometrium on day 14 of pregnancy, *PTGFR* mRNA abundance was downregulated (P-adjusted < 0.001; *p*-value = 0.016) and *SLC2A1* was upregulated (P-adjusted < 0.001; *p*-value = 0.001), while on day 19 *LIF* was upregulated (P-adjusted < 0.001; *p*-value =0.001) and *SLC1A1* was downregulated (P-adjusted < 0.001; *p*-value = 0.002). Other transcripts, *IL6ST*, *INSR*, *OXTR* and *SLC38A2*, which were all significantly upregulated on one or other day in the RNA-seq analysis, showed the same direction of change in expression after qRT-PCR; however the changes were not statistically significant (*p*-value > 0.05). We were not able to design usable primers to validate the changes of expression for *IGFBP3* and *KNG1*.

In conceptus samples, *H19*, *SLC2A3* and *SNRPN* transcripts showed no significant differences between conditions in the RNA-seq analysis, and this was confirmed by qRT-PCR. The RNA-seq analysis identified downregulation of *DNMT3B* and upregulation of *SLC1A4* on day 14 of synchronous pregnancy and upregulation of *NDN*, *PEG10* and *SLC7A5* on day 19 after synchronous ET, all of which was confirmed by qRT-PCR. Similarly, *SLC1A5* was downregulated on day 14 (P-adjusted and *p*-value < 0.005) and upregulated on day 19 (P-adjusted < 0.001; *p*-value = 0.029) in synchronous compared to asynchronous conceptuses, whereas *SLC2A5* was upregulated on day 14 (P-adjusted = 0.012; *p*-value = 0.002) and downregulated on day 19 (P-adjusted and *p*-value < 0.005) in synchronous membranes. The qRT-PCR results indicated upregulation of *IGF1* from day 14 of synchronous development (*p*-value = 0.007) whereas in the RNA-seq data, *IGF1* was only upregulated on day 19 (P-adjusted = 0.027; *p*-value = 0.082).

### 2.4. Cell Proliferation and Cell Attachment in Response to IGFBP3 Stimulation

Having identified *IGFBP3* as consistently upregulated in synchronous endometrium, we decided to investigate the potential function by examining the effect of IGFBP3 stimulation on the attachment and proliferation of trophectoderm cells, and compared its potency to a matrix component commonly used to promote cell attachment in culture, fibronectin. There was both a treatment and concentration effect on the mean number of cells attached per mm^2^ (Figure 7A). Fibronectin treatment promoted trophectoderm cell attachment in a concentration dependent fashion. More cells attached in the presence of 1000 ng/mL fibronectin than for 62, 125, 250 and 500 ng/mL (*p* < 0.0001). Cell attachment also increased between fibronectin concentrations of 125 and 500 ng/mL (*p* < 0.05). In the case of IGFBP3, a modest increase in cell attachment was observed between 62 ng/mL and 1000 ng/mL (*p* = 0.008). However, cell attachment was much higher in the presence of the highest fibronectin concentration (1000 ng/mL) than for BSA or IGFBP3 (*p* < 0.0001), but did not differ between IGFBP3 and BSA.

Cell proliferation was also measured after IGFBP3 stimulation, and in the presence of serum-containing versus serum-free medium (Figure 7B). Cell proliferation was higher in serum-containing medium (*p* < 0.05), whereas increasing the IGFBP3 concentration from 1 to 1000 ng/mL did not affect cell proliferation. After exposure to 10 or 100 ng/mL IGFBP3, cell proliferation was lower than in serum containing medium (*p* < 0.05).

### 2.5. Prostaglandin Release after Bradykinin Stimulation

One of the other genes consistently upregulated in synchronous endometrium was kininogen 1. To examine the possible role of endometrial kininogen, the effects of one of its major subcomponents, bradykinin, on prostaglandin secretion by cultured trophectoderm cells was examined. Release of PGE2, PGF2α and PGF1α from trophectoderm cells was augmented by the presence of bradykinin after both 6 and 24 h of culture. However, while increasing concentrations of bradykinin resulted in a dose-dependent increase in PGE2 release after 24 h (*p* < 0.0005; Figure 8), and there also was a tendency for PGF2α release at 24 h to increase with bradykinin dose (*p* = 0.055), there was no significant dose dependent increase of 6-keto PGF1α release. Overall, PGE2 was secreted in the highest concentrations (515–1486 pg/mL), followed by 6-keto PGF1 α (226–669 pg/mL) and PGF2α (56–202 pg/mL; data not shown).

## 3. Discussion

This study examined the effects of asynchronous embryo transfer on the transcriptome of the endometrium and conceptus membranes during early equine pregnancy, with the aim of discriminating the effects of progesterone from those of the presence of a conceptus on pathways involved in the establishment of pregnancy. Transcriptome comparison was performed between synchronous and asynchronous pregnancies on days 14 and 19 of conceptus development (6 and 11 days after transfer of a day 8 embryo), but also between day 14 synchronous and day 19 asynchronous pregnancies (day 14 endometrium in the presence of either a day 14 or a day 19 conceptus). Almost five times as many genes/transcripts were differentially expressed in conceptus membranes (3473; 25.5% of all detected transcripts) as in the endometrium (715; 4.6% of detected transcripts), with the number of differentially regulated genes increasing approximately 5 fold between days 14 and 19 in the conceptus membranes but only modestly in the endometrium. These differences presumably reflect the rapid and extensive changes occurring in the conceptus membranes at this stage of development, many of which will help to establish and enhance nutrient transfer to the embryo and dialogue with the endometrium; the large number of DEGs in the conceptus membranes also illustrates the dynamism and relative plasticity of the extra-embryonic membranes during the preimplantation period and the profound effect of 5 days of negative uterine asynchrony on conceptus development [40].

The changes identified by RNA-seq were broadly replicated, and thereby validated, by qRT-PCR, although the statistical significance of the changes was more consistent for the genes selected in the conceptus membranes than those examined in the endometrium. In the endometrium, although the directions of change in expression were the same for RNA-seq and PCR data, the latter were less likely to reach statistical significance. In this respect, endometrial biopsies were collected from different mares and biological noise due to genetic variability between mares may have reduced consistency [14]. Moreover, tissue recovered by endometrial biopsy is composed of many different cell types, and it is not possible (except by laser-capture microdissection) to ensure identical cell composition between biopsies. This could, in part, explain the small differences between the RNA-seq and the PCR results observed in endometrial samples. The conceptus membranes are likely to be more uniform because of the small number of cell types present and the relatively well defined temporal pattern of development.

We previously reported that negatively asynchronous ET delays conceptus development at the level of the whole conceptus and for selected genes known to exhibit parent-of-origin imprinting [40] or related to nutrient uptake [41,42]; the retardation in development presumably results from differences in histotroph composition due to the reduced duration of endometrial exposure to P4 prior to embryo introduction. The transcriptome comparisons performed between conceptus membranes at the different stages (D14-syn/asyn; D19-syn/asyn) indicated distinct profiles of DEGs, but gene ontology analysis revealed similar overrepresented categories to those reported previously between day 10, 12 and 14 compared to day 8 conceptuses by Klein and Troedsson [19]. Indeed, Klein and Troedsson reported that the cellular compartment ‘nucleus’ and the biological processes ‘RNA processing’, ‘regulation of gene expression’ and the molecular function ‘RNA polymerase activity’ were downregulated in older conceptuses. We similarly found that the biological processes ‘translation’, and cellular components and molecular functions related to the nucleus were downregulated in synchronous compared to asynchronous (i.e., retarded) day 14 conceptuses. While Klein and Troedsson reported genes involved in positive regulation of the immune system to be downregulated in older conceptuses [19], we found that genes clustered under antiviral defense and various aspects of activation of the immune response were overrepresented among downregulated genes on day 19, which supports the idea that, as it develops, the semiallogenic conceptus becomes increasingly capable of hiding its immunogenicity or suppressing the uterine response to its foreign alleles. Functional categories that were overrepresented among the upregulated genes in synchronous conceptuses on day 14 of development included positive regulation of exocytosis and peptide hormone secretion. Similarly, Klein and Troedsson [19] reported upregulation of transcripts involved in the response of the conceptus to extracellular stimuli including hormones. On day 19 of development, synchronous conceptuses showed enrichment of transcripts for the biological processes ‘cell adhesion’, ‘cell migration’ and ‘inflammatory responses’, which undoubtedly reflect the developmental changes that occur in the membranes during the formation of the placenta and in the preparation for implantation. Most striking among the potential mechanisms of signaling and communication, were the extremely large numbers of transcripts relating to ‘extracellular exosome’ that were differentially regulated at both day 14 and, in particular, day 19. Recent studies have demonstrated release of extracellular vesicles by both conceptus and uterus during early pregnancy in the ewe [43]; these vesicles were enriched for both specific proteins, mRNAs and miRNAs [44]. Since extracellular vesicles from both conceptus and uterus were taken up by trophectoderm and endometrial epithelial cells but were not found in deeper tissues, they were proposed to play a role in embryo-maternal communication. That the vesicular cargo in the uterus of pregnant ewes appears to be differentially regulated by progesterone [44] may explain why we saw differential (both up and down) regulation of such large numbers of genes related to extracellular exosome and exocytosis following asynchronous ET. Most importantly, it suggests that exosomes and other extracellular vesicles secreted by equine endometrium and conceptus membranes may play important roles in equine embryo-maternal communication during the periods of MRP and preparation for implantation. It would therefore be worthwhile examining both conceptus and endometrial extracellular vesicles to see how their protein, mRNA and miRNA cargos vary temporally or in response to progesterone or conceptus presence (endometrium), and investigating how these components may contribute to embryo-maternal communication and the establishment of pregnancy in the mare.

In endometrial tissues, we found more than five hundred DEGs between the synchronous and asynchronous conditions on days 14 and 19 of conceptus development, but only fourteen DEGs in the D14-syn/D19-asyn comparison. This indicates that the effect of duration of P4 exposure on the endometrial transcriptome is more profound than that of the duration of conceptus presence; this further implies that progesterone and/or the straightforward absence or presence of a conceptus plays the dominant role in regulating endometrial gene expression during the establishment of pregnancy, with the precise age and duration of presence of the conceptus playing a more subtle role in refining endometrial gene expression. Of course, in the groups D14-syn/asyn and D19-syn/asyn, we compared endometrium at different stages after ovulation, namely day 14 versus day 9 and day 19 versus day 14 in the presence of a day 14 or 19 conceptus, respectively. Not surprisingly, this resulted in large numbers of DEGs, namely 523 on day 14, of which 413 were specific to the D14-syn/asyn comparison, and 715 DEGs on day 19, of which 600 were specific to the D19-syn/asyn comparison. Gene ontology analysis revealed various over-represented functional terms among the up- and downregulated genes, with some of them previously shown to be enriched in equine endometrium simply as an effect of increased time after ovulation or the presence of a conceptus [14,18]. The functional terms found to be upregulated during pregnancy that were shared by our study and previous studies, include ‘calcium and peptide binding’, ‘cell-cell signaling’, ‘regulation of cell growth’, ‘negative regulation of apoptotic process’, ‘membrane’ and ‘extracellular exosome’ [14,18]. That the endometrium responds to the presence of a conceptus from fairly soon after the latter’s arrival in the uterus is indicated by changes in oxytocin sensitivity from around day 10 after ovulation [8,10,11]. In the present study, gene ontology further suggests a remodeling of the endometrium from at least day 14 of gestation, as indicated by upregulation in genes for cell growth and migration, which is presumably accompanied by changes in the overlying glycocalyx as reflected by enhanced expression of genes for proteinaceous extracellular matrix; the extracellular matrix changes could be involved in modification of the endometrial surface to prevent conceptus adhesion before fixation, but then to facilitate conceptus–endometrium adhesion and the preparation for implantation following vesicle fixation on day 16. Alternatively, some of the changes related to the extracellular matrix may be involved in the changes in the size and composition of the blastocyst capsule during this period [4,45], which is known to involve an endometrial contribution and to incorporate uterine derived proteins [4,45,46]. Interestingly, as discussed for the conceptus membranes, it appears that exosomes and peptide binding are likely to play important, but to date incompletely understood, roles in the conceptus–endometrium interaction [43]. Angiogenesis and vascularization of the endometrium also appears to be an important event during the preparation for implantation, as evidenced by the enrichment of the cellular components ‘blood microparticles’, ‘inflammatory response’ and ‘cellular response to vascular endothelial growth’ on day 19 of synchronous pregnancy, and supported by the detection of VEGF and its major receptors in the endometrium during both the estrous cycle and throughout gestation [47].

In the D14-syn/D19-asyn comparison, the two states are linked by including a recipient mare/endometrium on day 14 after ovulation, and distinguished by the presence of a conceptus for a longer or shorter duration (6 or 11 days after ET, corresponding to day 14 and day 19 of conceptus development). Only six genes were uniquely differentially expressed between D14-syn and D19-asyn endometrium, three of which were upregulated by the presence of an older conceptus for a longer duration (i.e., in D19 asynchronous condition: *MGLL*, *LIPG* and *IL36G*). The gene *MGLL* codes for monoglyceride lipase and was previously found to be upregulated during early equine pregnancy [14]. MGLL is a serine hydrolase that catalyzes the conversion of monoacylglycerides to free fatty acids and glycerol; the upregulation of *MGLL* in the presence of a more advanced embryo could be explained by increasing embryonic demands for free fatty acids and glycerol, i.e., to improve the availability of lipid building blocks. While fatty acids and glycerol are known to cross the human placental barrier and be used by the fetus [48], during early equine pregnancy they presumably need to be carried or transported into the conceptus. In this respect, the progesterone-dependent endometrial protein uterocalin (P19) is specifically present during early pregnancy, is able to bind a variety of fatty acids and has been proposed to act as a carrier for the fatty acids required to support embryonic development [49]. *LIPG*, also known as endothelial lipase, also has substantial phospholipase activity and could play an additional role in endometrial lipid metabolism to help provide fatty acids and lipid building blocks to the conceptus. The interleukin 36 gamma gene (*IL36G*) is part of the interleukin 1 (IL1) cytokine family. During decidualization of human endometrial stromal cells, cell receptivity to the IL1 family is modulated [50], suggesting a role in endometrial receptivity. Endometrial expression of these three genes is stimulated by the presence of a more advanced conceptus, presumably as a result of specific factors expressed by the conceptus that help modulate the uterine environment to better support the development of a more advanced conceptus.

Interestingly, we identified three genes, *KNG1*, *IGFBP3* and *FAM118B*, that were differentially expressed in all endometrial comparisons, D14-syn/asyn, D19-syn/asyn and D14-syn/D19-asyn. In particular, *IGFBP3* and *KNG1* were upregulated in the endometrium of synchronous pregnancies on days 14 and 19, but also in day 14 endometrium in the presence of a day 14 compared to a day 19 conceptus. *FAM118B* had the opposite profile of expression change to *KNG1* and *IGFBP3*. Therefore, rather than being upregulated simply by the presence of an older conceptus or duration of exposure to progesterone, *IGFBP3* and *KNG1* expression seemed to reflect appropriate synchrony between uterus and conceptus. Moreover, the fact that these genes were differentially expressed in all three comparisons suggests a role for matching endometrial stage to embryo development during the establishment of pregnancy, with effects on the conceptus either directly or by altering other aspects of endometrial function.

KNG1 (kininogen 1) is part of the kinin–kallikrein system. Alternative splicing of *KNG1* generates two proteins, a high molecular weight kininogen (HMWK) and a low molecular weight kininogen (LMWK). Kallikrein (*KLK*) encodes a serine protease, kallikrein. The precursor prekallikrein is activated by Factor XII and subsequently cleaves HMWK to release bradykinin (BK). Bradykinin is a proinflammatory peptide that acts via specific receptors, bradykinin receptors B1 and B2 (BKRB1 and BKRB2), and is involved in blood pressure regulation and in inflammatory responses that increase vascular permeability, to cause vasodilatation, and regulate prostaglandin production [51,52,53]. Various members of the kinin–kallikrein system have previously been detected in equine, human, murine and porcine endometrium or conceptuses. For example, bradykinin protein was found in porcine ULF, with expression increasing during pregnancy [54], while LMWK, BKRB2, Factor XII and plasma kallikrein (KLKB1) are all expressed in porcine endometrium [51,54]. Factor XII has been detected in human decidual tissues [55] and in day 12 equine conceptuses [19]. KNG1 has been detected in the endometrium, and reported to be upregulated during early equine pregnancy [14,18], and various tissue kallikreins (KLK1, KLK3, KLK4 and KLK14) are expressed by equine endometrium and the conceptus [18,19,20]. Furthermore, the kininogen protein has been detected in equine uterine luminal fluids on day 13 after ovulation, although it was not present in yolk sac fluid [22], and ULF kininogen concentration increases as pregnancy progresses in the pig [54]. These results suggest that the kinin–kallikrein system is active during early pregnancy and could be involved in the establishment of pregnancy by stimulating an inflammatory response, angiogenesis and prostaglandin production in the endometrium and/or the conceptus [51,54,55,56,57]. The horse embryo is known to secrete large amounts of prostaglandins, including PGE_2_, PGF_2α_ and PGI_2_, which are thought to play a role in stimulating a local increase in vascularity and in conceptus mobility [58]. To examine whether KNG1 may be involved in conceptus prostaglandin production, we stimulated primary monolayers of trophectoderm cells with different concentrations of bradykinin for 6 and 24 h. Production of PGE_2_, PGF_2α_ and 6-keto PGF_1α_ (the stable metabolite of PGI_2_) increased between 6 and 24 h of culture with bradykinin and, by 24 h, concentrations of all three were significantly higher than after control (6 h no bradykinin) cultures. PGE_2_ and PGF_2α_ showed evidence of a dose response to the concentration of bradykinin, but release of all prostaglandins was enhanced by the lowest concentration of bradykinin tested. Bradykinin is known to induce the production of specific ranges of prostaglandins, dependent on the tissue stimulated, e.g., PGE_2_ in gastric mucosa [59], and PGE_2_, PGF_2α_ and 6-keto PGF_1α_ in ovarian follicles [60,61]. Since the bradykinin receptors, B1 and B2, were not expressed in the conceptus membranes, bradykinin must work via an alternative pathway to stimulate prostaglandin production in trophectoderm. In pulmonary smooth muscle cells, which secrete many cytokines and growth factors, bradykinin has been shown to induce prostaglandin production by upregulating prostaglandin-endoperoxidase synthase 2 (PTGS2 also known as COX2) [62,63]. The latter studies also provided evidence that bradykinin induced PTGS2 via the CRE-binding site of the PTGS2 promotor. It seems reasonable to speculate that, in equine trophectoderm cells, bradykinin stimulates PTGS2 expression and thereby increases the synthesis of at least PGE_2_ and PGF_2α_. PTGS2 metabolizes arachidonic acid into prostaglandin H2 (PGH_2_), which is a common substrate for the production of other prostaglandins. Prostaglandin production by the equine conceptus is thought to be required for conceptus mobility in the uterine cavity by inducing myometrial contractions [58], most likely due to PGE2 and PGF_2α_, which are both secreted by the conceptus [9]. The sheep conceptus is also known to secrete PGE_2_, PGF_2α_ and PGI_2_, which act in a paracrine fashion on the endometrium to regulate the expression of genes associated with endometrial remodeling, cellular function (cell migration, proliferation and attachment), angiogenesis and nutrient transport [64]. Thus, endometrial expression of KNG1 and the release of bradykinin could have important effects on prostaglandin production, and may have further downstream effects on functions important to conceptus development and the maintenance of pregnancy, such as angiogenesis and nutrient transport.

IGFBP3 (insulin like growth factor binding protein 3) is part of the IGF-system and has a high affinity for IGF1 and IGF2. Binding of IGF1 or 2 to IGFBP3 can increase their half-life in the circulation, but can also reduce their availability to bind to receptors. Furthermore, IGFBPs have been reported to have direct effects such as the stimulation of cell proliferation and cell attachment, although the mechanisms of these effects are not well understood [65]. IGFBP3 is expressed in the uterine lumen during early pregnancy in ruminants [65,66], and is progesterone-regulated in sheep [26]. In the horse, IGFBP3 protein has been detected in association with the blastocyst capsule, in the ULF and in medium conditioned by conceptus membranes [67]. Moreover, *IGFBP3* is expressed in conceptus membranes and is upregulated in the endometrium of pregnant compared to non-pregnant mares [14,18,67]. However, very little is known about the function of IGFBP3 during early pregnancy, other than the assumption that it is likely to regulate IGF bioavailability, where IGF1 is known to be produced by both conceptus and endometrium during early pregnancy in the mare [68]. On the other hand, it has also been postulated that IGFBP3 influences cell proliferation and cell survival directly [69,70]. Similar to what has been described for IGFBP1 in ovine trophectodermal cells [65], we tested the effect of IGFBP3 on proliferation and attachment to plastic of equine trophectoderm cells. Contrary to expectations, none of the assays performed showed a clear effect of IGFBP3 on trophectoderm cell behavior. Precoating tissue culture wells with IGFBP3 did not enhance cell attachment to anywhere near the extent that fibronectin coating did. Addition of IGFBP3 to the culture medium did not enhance cell proliferation, although neither was it detrimental in that cell proliferation indices were no different to those in the control condition. Perhaps endometrial IGFBP3 serves primarily to regulate bioavailability of IGFs to the equine conceptus; alternatively, IGFBP3 may have predominantly autocrine or paracrine effects on endometrial cells to help prepare the uterus for implantation.

*FAM118B* has recently been identified as a component of Cajal bodies (coiled bodies) in the nucleus that are involved in the biogenesis of small ribonucleoproteins. *FAM118B* plays an active role in Cajal body formation in that it is required for their assembly, splicing and cell proliferation [71]. Very little is known about the ubiquity or function of Cajal bodies during early pregnancy.

This study has generated lists of genes, in both endometrium and conceptus membranes, expression of which is influenced by the duration of exposure of the endometrium to progesterone, age of the conceptus and/or duration of conceptus presence. Clearly, length of progesterone exposure and the presence of the conceptus combine to regulate the endometrial transcriptome. Circulating progesterone and factors secreted by the conceptus affect groups of genes related to specific biological processes in the endometrium, which in turn affect the conceptus membranes, thereby creating an interrelationship that should synchronize development of the two and optimize preparation for implantation and the formation of a placenta. Significant asynchrony between the conceptus and the uterine environment, in the form of a uterus lagging behind the developmental stage of the conceptus, leads rapidly to a marked delay in conceptus development. This demonstrates a very precise interaction or dialogue between the conceptus and the endometrium, and indicates that the equine conceptus is able to sense the endometrial stage and delay its development accordingly. The exact mechanisms by which the conceptus senses the uterine stage are not known, but it is most likely due to the absence, or inappropriate concentrations, of specific factors expressed or secreted by the endometrium. Moreover, the role of extracellular vesicles as a mode of communication between conceptus and endometrium is worthy of further investigation. Finally, identifying the stimulatory effect of bradykinin on prostaglandin release by equine trophectoderm cells is an interesting step in understanding the regulation of conceptus PG production during early equine pregnancy. Expression of KNG1 by the endometrium could be primarily involved in stimulating PG production by the conceptus, but may also have other functions. Further studies will be needed to understand better the roles of bradykinin and IGFBP3 in matching conceptus and endometrial stage, but the very fact that optimal embryo-uterine asynchrony consistently upregulates their expression indicates that they are likely to play an important role in the intricate interaction between the endometrium and the conceptus during the establishment of pregnancy and preparation for implantation.

## 4. Material and Methods

### 4.1. Animals and Tissue Collection

All animal procedures were approved by the Utrecht University’s Animal Experimentation Committee (permit number 2012.III.02.020). Material for the transcriptomic analysis was collected from 22 warmblood mares, and used in previous studies [40,41,42]; material for the in vitro experiments was harvested from an additional 5 mares. Mares were between 4 and 15 years of age and maintained on pasture with ad libitum access to grass, hay and water. During estrus, the mares were monitored every other day by transrectal ultrasonography (MyLab30 ultrasound machine equipped with a 7.5 MHz linear transducer; Esoate, Maastricht, The Netherlands) [72]. When the dominant follicle exceeded 35 mm in diameter, mares were inseminated with a minimum of 500 × 10^6^ sperm cells from a single fertile stallion, and thereafter examined daily until ovulation was observed; insemination was repeated every second day until ovulation. Mares used as recipients for ET were monitored similarly to determine the ovulation date, but were not inseminated. For the ET experiment, the degree of synchrony or asynchrony between the donor and recipient mares was managed by manipulating the estrous cycle using a PGF2α analogue (37.5 µg D-cloprostenol; Genestranvet: Eurovet Animal Health B.V., Bladel, The Netherlands) to induce luteolysis, and human chorionic gonadotrophin to induce ovulation (hCG; 1500 i.u. Chorulon: Intervet, Boxmeer, The Netherlands).

### 4.2. Embryo Collection and Transfer

For the ET experiment, blastocysts were recovered from donor mares by uterine lavage 8 days after ovulation, as described previously [73]; recovered embryos were washed in holding medium (Syngro; Bioniche Animal Health, Pullman, WA, USA). In total, 20 pregnancies were established after transfer of day 8 embryos (grade 1-2) to recipient mares that had either ovulated on the same day as the donor (synchronous; *n* = 10), or 5 days after the donor mare (asynchronous; *n* = 10). Conceptuses were then recovered on day 14 or day 19 of development (*n* = 5 per group; Figure 9). Day 14 conceptuses were recovered 6 days after ET by uterine lavage via a sterile endotracheal tube, and day 19 conceptuses were recovered 11 days after ET using an endoscopically-guided net, after puncture of the membranes and aspiration of the yolk-sac fluid using a sharpened PTFE catheter [72]. Following conceptus recovery, an endometrial biopsy was collected, using alligator forceps (141965; Jørgen Kruuse A/S, Langeskov, Denmark), from the base of one or another uterine horn for day 14 conceptuses (mobile phase), or from the site of conceptus apposition for day 19 conceptuses (fixed location; *n* = 5 per group) under video-endoscopic guidance. For conceptus membrane culture, pregnancy status was monitored by transrectal ultrasonography from day 14 after ovulation, and conceptuses were recovered on day 19 by uterine lavage via a sterile endotracheal tube (*n* = 5 conceptuses).

Conceptuses and endometrial biopsies were washed with sterile 0.9% NaCl. Under a stereomicroscope (Olympus SZ-60: Olympus Nederland B.V., Leiderdorp, the Netherlands), the blastocyst capsule was removed from the conceptuses and the embryonic disc region, or embryo proper, were dissected from the conceptus membranes using microsurgical scissors. On day 14, conceptus membranes consisted primarily of bilaminar yolk-sac (endoderm and trophectoderm) with a relatively small trilaminar area. Within day 19 conceptuses, mesoderm formation had progressed appreciably, resulting in trilaminar membrane over the bulk of the conceptus. The bilaminar and trilaminar yolk-sac regions of day 19 conceptus membranes were not separated for analysis. Yolk-sac membranes from ET conceptuses were snap frozen in liquid nitrogen and stored at −80 °C, whereas the conceptus membranes used for culture were kept in warm PBS prior to culture.

### 4.3. RNA Extraction, DNA Library Preparation and Sequencing

Total RNA was extracted using the AllPrep DNA/RNA/Protein Mini kit (Qiagen, Venlo, The Netherlands). Endometrium and conceptus membranes (20–40 mg) were homogenized in 600 µL lysis buffer and total RNA was eluted with 40 µL RNase-free water, and treated with DNAse I for 30 min at 37 °C and 10 min at 65 °C (1 IU/μg; RNase-free DNase kit; Qiagen). Total RNA quantity and quality was determined respectively by spectrometry (NanoDrop ND 1000; Isogen Life Sciences, De Meern, The Netherlands) and using a BioAnalyzer 2100 (Agilent Technologies, Palo Alto, CA) with a RNA 6000 Nano Chip, according to the manufacturers’ instructions. The best 4 samples from each group were selected (RNA integrity number (RIN) values >8.0 for conceptus membrane samples and >6.6. for endometrial samples), and DNA library preparation was performed using the Illumina Truseq Stranded Total RNA low-throughput RNA Sample Prep Kit (Illumina, Inc.; San Diego, CA, USA) with 500 ng of total RNA, following depletion of ribosomal RNA using the Ribo-Zero Human/Mouse/Rat kit. DNA library quality was controlled using a BioAnalyzer 2100, the Agilent DNA 1000 and the Qubit assay (Agilent Technologies). Next, bar-coded endometrial samples and conceptus membrane samples were pooled separately. The sequencing was performed at the Utrecht Sequencing Facility on an Illumina Next Seq 500 (San Diego, CA, USA) with a single-read mode and a read length of 75 bases.

### 4.4. RNA-seq Analysis and Bioinformatics

The Fastq files were analyzed using a locally installed version of Galaxy [74]; the reads were cleaned and trimmed using Trimmomatic (v0.33) using the following parameters: 3’-end trimming with sliding window (size = 4, average quality score ≥28), headcrop at the 5’-end (number of bases = 3) and minimum length read (50 bases). The reads were aligned to the horse genome (equuCab2) using Tophat (v2.0.11, splice junction mapper for RNA-seq reads [75]). Next, the number of reads mapped to the exon region of each gene was evaluated using QuasR (v1.8.2) qCount [76] and only the genes with more than 10 counts were used for statistical analysis. Differentially expressed genes (DEG) were detected using the DESeq2 tool (v1.8.1) in R [77] and we compared d14 syn vs. d14 asyn, d19 syn vs. d19 asyn and day 14 syn vs. day 19 asyn. Significance thresholds were set as follows: false discovery rate (FDR) <5% and a log2 fold change (FC) greater than 1.4 or lower than −1.4 (equivalent to a fold change of 0.49 or −0.49) and P-Adjusted less than 0.04. Gene ontology clustering was performed using the Database for Annotation, Visualization and Integrated Discovery (DAVID, v6.8) using the functional annotation chart tool [78].

### 4.5. Quantitative RT-PCR

To validate the sequencing results, RT-qPCR was performed for a number of genes for which we already had validated primers. A selection was made per tissue (conceptus and endometrium) to include genes that were upregulated, downregulated and not altered by asynchrony at one or both time points. RT-qPCR was performed using the same RNA samples as used for sequencing. Total RNA (1 µg) was reverse-transcribed to cDNA using Superscript III (Invitrogen) followed by qRT-PCR as described previously [40]. Primers were designed using PerlPrimer 1.1 (http://perlprimer.sourceforge.net/); where multiple splice variants existed for a target gene, the primers were designed for areas of the mRNA that were conserved across the variants. The primers were produced at Eurogentec (Seraing, Belgium) and specificity was tested by DNA sequencing (ABI PRISM 310 Genetic analyzer; Applied Bio-System, Foster City, CA, USA). Real-time PCR was carried out in 15 µL of reaction mix including 7.5 µL of IQ SYBR^®^ Green Supermix (BioRad; Veenendaal, The Netherlands), 0.5 mM of primer (forward and reverse; Appendix A) and 1 µL of cDNA, on an IQ5 Real-Time PCR detection System (BioRad). For each gene, a melting curve was performed to verify product specificity, and a 10-fold serial dilution of the target gene PCR product was amplified simultaneously with the samples to establish a standard curve that was used to quantify sample mRNA expression. Relative gene expression was expressed as the ratio of target gene expression to the geometric mean for the housekeeping genes GAPDH, HPRT1 and SRP14 for conceptus samples, and GAPDH and SRP14 for endometrial samples. Finally, the log2 fold change between the synchronous and asynchronous samples was calculated for day 14 and day 19 of pregnancy, and an independent *t*-test was performed.

### 4.6. Culture of Conceptus Membrane Cells

Conceptus membranes were washed 3 times in PBS and then transferred into M199 medium (M199-22340 with HEPES; ThermoFisher Scientific, Paisley, UK) supplemented with 1 mg/mL Collagenase A (Sigma-Aldrich; Zwijndrecht, the Netherlands) and 0.1% P/S (penicillin = 10,000 IU/mL and streptomycin = 10,000 µg/mL; Sigma-Aldrich), the membranes were cut into small pieces using a razor blade and passed through sterile 18 and 21-gauge needles to further disrupt cell–cell attachment. In total, the membranes were maintained for 30 min at 37 °C in collagenase containing medium. The cells were then washed twice in standard culture medium (M199 supplemented with 20% fetal bovine serum and 0.1% P/S) and centrifuged at 500× *g* for 5 min. Finally, the cells were pslated in T-25 flasks with standard culture medium for 3 days at 37 °C in a humidified atmosphere containing 5% CO_2_-in-air. The culture medium was replaced every other day, and the cells were used for in vitro assays after 1 or 2 passages. All trophectoderm function experiments were performed on cells obtained from three different conceptuses, with each condition performed in triplicate.

### 4.7. Cell Attachment Assay

Culture medium was replaced by the serum-free medium (M199 with 0.01% P/S) for 24 h. A 24-well plate containing a glass coverslip in each well was coated overnight at 4 °C with 450 µL of the following proteins: BSA (Sigma-Aldrich: 1000 ng/mL in PBS), fibronectin from bovine plasma (F1141, Sigma-Aldrich: 62, 125, 250, 500 and 1000 ng/mL in PBS) or recombinant human IGFBP3 (10430H07H50, Sino Biological Inc., Beijing, China: 62, 125, 250, 500 and 1000 ng/mL in PBS), where IGFBP3 was selected on the basis of the RNA-seq results. Next, non-specific binding to the wells was blocked with heat-denatured BSA (10 mg/mL in PBS) for 1 h at room temperature (RT). Freshly trypsinized cells (*n* = 47,000 cells in 500 µL serum-free media) were then added to each well and allowed to attach for 2 h (37 °C, 5% CO_2_). After removing the medium, the wells were washed with PBS to remove any unattached cells, and attached cells were fixed using methanol:acetone (1:1) for 20 min at RT. Cell nuclei were stained with DAPI (1:50 in PBS: Molecular Probes, leiden, the Netherlands) for 30 min at RT in the dark. Coverslips were mounted with Vectashield (Vector Laboratories, Burlingame, CA, USA) and 5 non-overlapping sections were imaged using a fluorescence microscope (Olympus BX60; Olympus Nederland B.V., Leiderdorp, the Netherlands) equipped with a Leica DFC425C camera (Leica, Wetzlar, Germany) to count the number of cells attached per mm^2^. The effect of the different conditions on cell attachment was examined by one-way ANOVA (*p* < 0.05). 

### 4.8. Cell Proliferation Assay

In a 24-well plate containing glass coverslips, freshly trypsinized cells were allowed to attach for 6 h (*n* = 30,000 cells in 500 µL standard culture medium) before replacing the medium with 500 µL serum-free medium for 24 h. Next, cells were treated with different concentrations of IGFBP3 (1, 10, 100 and 1000 ng/mL in 500 µL serum-free medium), 500 µL serum-free medium or with 500 µL normal culture medium (20% FBS, 0.1% P/S, M199). At the same time, BrdU (5-Bromo-2′-deoxyuridine; B5002, Sigma-Aldrich; final concentration: 10 µM) was added to each well, and the cells were then cultured for 24 h at 37 °C in a humidified atmosphere of 5% CO_2_-in-air. Following incubation, the cells were rinsed with PBS and fixed using methanol:acetone (1:1) for 20 min at RT. The cells were then treated with HCl (2M) for 20 min at RT, followed by a blocking step (PBS/BSA 0.5%) for 10 min. Finally, the cells were stained with anti-BrdU (FITC mouse anti-BrdU; BD Biosciences, San Jose, CA: diluted 1:50 in PBS) for 2 h at RT and with DAPI (1:50 in PBS) for 30 min at RT. Coverslips were mounted with Vectashield and 5 non-overlapping sections were imaged using a fluorescence microscope (Olympus BX60 with a Leica DFC425C camera) to count both the total cell number (DAPI) and the number of proliferating cells (BrdU positive) per mm^2^. The effects of culture medium and IGFBP3 concentration on cell proliferation were analyzed by two-way ANOVA (*p* < 0.05).

### 4.9. Bradykinin Stimulation

Freshly trypsinized cells were seeded into 24-well plates (*n* = 30 000 cells in 500 µL normal culture medium) and incubated for 24 h (37 °C, 5% CO_2_). The medium was replaced by serum-free medium and, after 24 h, cells were treated with 500 µL bradykinin (catalogue number B3259-1MG, Sigma-Aldrich: 0.5, 1, 10 and 100 ng/mL in M199) for 6 h and 24 h, or with 500 µL serum-free medium for 6 h (negative control). Culture medium was collected at the appropriate times and stored at −80 °C for subsequent prostaglandin (PG) concentration analysis.

### 4.10. Prostaglandin Assay

PGs released from cultured conceptus membrane cells were assayed using high-performance liquid chromatography-tandem mass spectrometry (HPLC-MS/MS), as described previously [79,80]. A set of standard samples was prepared by adding stock PG solution to serum-free medium (M199, 0.1% P/S) to obtain final concentrations of 6.25, 12.5, 25, 50, 100 and 200 pg/µL. An internal standard (IS; 16,16-dimethyl-PGF2α, 40 pg/µL) was added to each standard sample (100 µL per sample). Chromatograms for standards were used to establish characteristic retention times (RTs) for each compound, while the calibration lines were used to verify that the MS signal was linear for all analytes over this range. The peak-area ratios of each analyte to the IS were calculated and plotted against the concentration of the calibration standards. Calibration lines were calculated using the least squares linear regression method.

Extraction of PGs from (trophectoderm cell conditioned) culture medium was performed as described previously [80]. Culture medium was thawed, and 200 µL was mixed with 100 µL of IS, 200 µL of ammonium acetate (0.2 M, pH 3.3) and 800 µL of ethyl acetate (0.0002% butylated hydroxytoluene) by vortexing for 1 min. The mixture was then centrifuged at 12,100× *g* for 5 min and chilled to −80 °C. After 60 min, the upper ethyl acetate layer was pipetted into a new tube. The ethyl acetate extraction procedure was repeated once more. The pooled samples were then placed in a Speedvac system (ThermoFisher) until the solvent was completely evaporated; the residue was dissolved in 40 µL of 30% methanol. Thirty µL of each sample was transferred to a polypropylene microwell plate and stored at −20 °C until HPLC-MS/MS analysis was performed.

For each sample, the ratio of the peak area of the internal standard to the peak area for PGE2, PGF2α and 6-keto PGF1α was measured in pg/mL. For all samples, we measured the log2 fold change of the ratio of bradykinin stimulation to the negative control. The effects of the duration of treatment (time) and the concentration of bradykinin (stimulation) used on PGE2, PGF2α and 6-keto PGF1α release were tested using a linear mixed model (R Studio: Boston, MA, USA)). Time and stimulation were used as fixed factors, individual embryo culture was included as a random effect, and we included a nested random effect of stimulation within time. An ANOVA was used to examine whether the values of the estimates obtained in the linear mixed model were significant (*p* < 0.05).

## Figures and Tables

**Figure 1 ijms-21-02562-f001:**
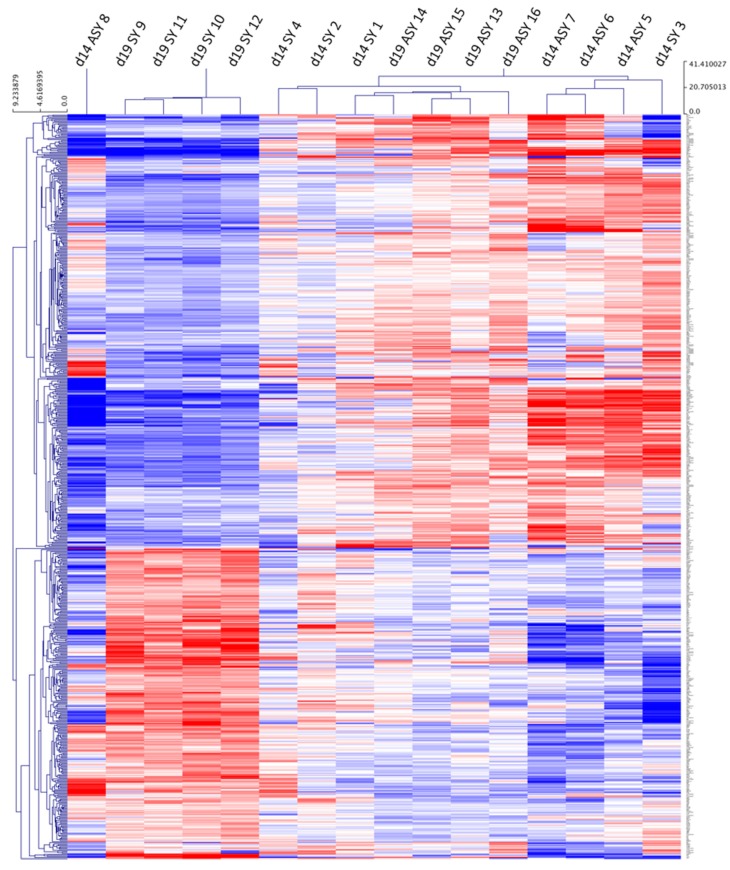
Hierarchical cluster analysis of differentially expressed genes (DEGs) with a log2 fold change higher than 1.4 or lower than -1.4 for equine endometrial samples collected after transfer of day 8 horse embryos to synchronous (day 8: SY) or asynchronous (day 3: ASY) recipient mares. Endometrial samples were collected on day 14 (d14) or 19 (d19) of embryo development (blue = downregulated genes; red = upregulated gene). The sample number is indicated by a number from 1 to 16.

**Figure 2 ijms-21-02562-f002:**
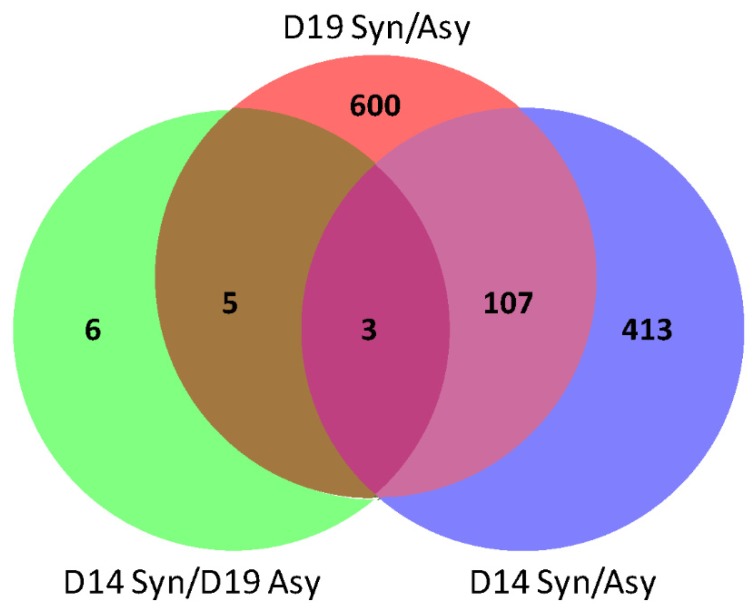
Venn diagram for the overlap of DEGs in the endometrium among three pair-wise comparisons, day 14 of conceptus development between synchronous (day 14 endometrium) and asynchronous (day 9 endometrium; D14 syn/asyn), on day 19 of conceptus development between synchronous (day 19 endometrium) and asynchronous (day 14 endometrium; D19-syn/asyn) and between day 14 synchronous and day 19 asynchronous pregnancies (D14-sync/D19-asyn). The numbers of DEGs in each compartment is depicted.

**Figure 3 ijms-21-02562-f003:**
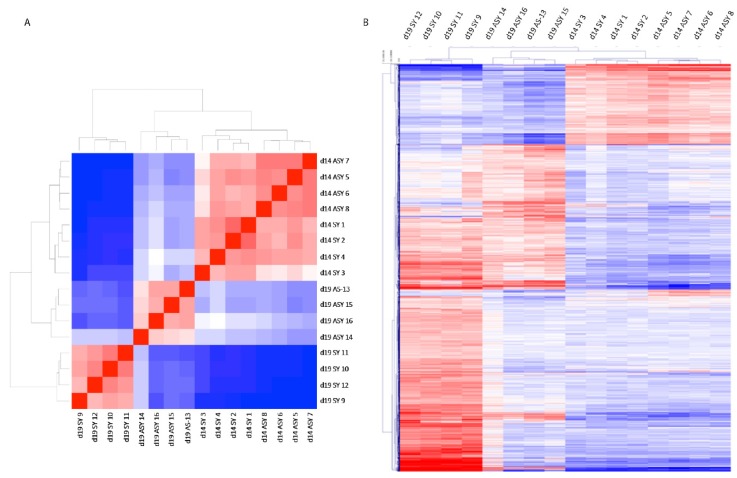
Pair-wise correlation heat map (**A**) and hierarchical cluster (**B**) analysis of DEGs with a log2 fold change higher than 1.4 or lower than -1.4 for equine conceptus membrane samples recovered after transfer of day 8 embryos to a synchronous (day 8: SY) or asynchronous (day 3: ASY) recipient mare, and collected on day 14 (d14) or 19 (d19) of conceptus development (blue = downregulated genes; red = upregulated genes). The sample number is indicated by a number from 1 to 16.

**Figure 4 ijms-21-02562-f004:**
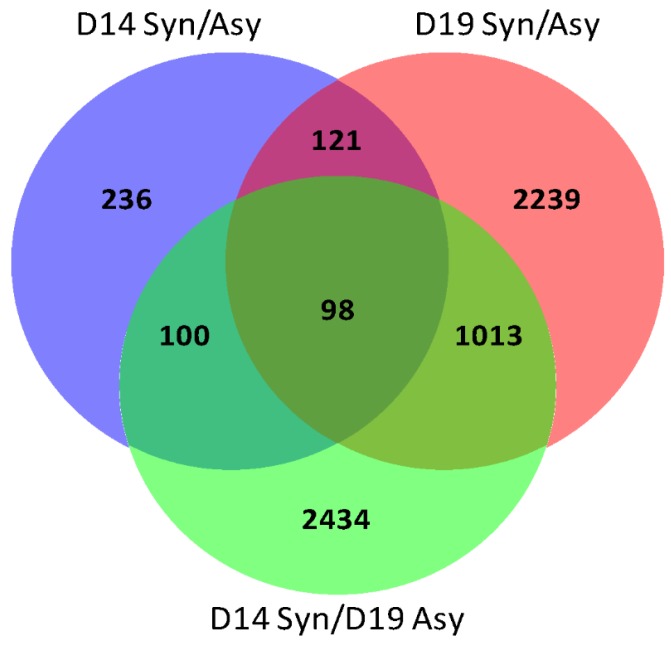
Venn diagram for the overlap of DEGs in equine conceptus membranes between three pair-wise comparisons, day 14 of pregnancy between synchronous (day 14 recipient) and asynchronous (day 9 recipient; D14 syn/asyn) pregnancies, on day 19 of pregnancy between synchronous (day 19) and asynchronous (day 14 recipient; D19-syn/asyn) and between day 14 synchronous day 19 and asynchronous (D14-sync/D19-asyn). The numbers of DEGs in each compartment is depicted.

**Figure 5 ijms-21-02562-f005:**
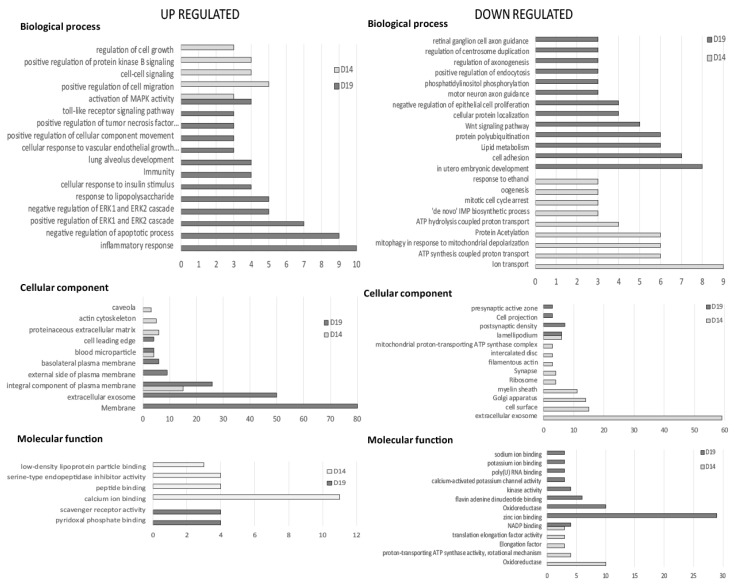
Gene ontology analysis for genes upregulated and downregulated in equine endometrium on day 14 and 19 of pregnancy between synchronous (recipient and donor ovulated on same day) and asynchronous (recipient ovulated 5 days after the donor) embryo transfer (ET; D14 syn/asyn and D19 syn/asyn). The major categories within the biological process, cellular component and molecular function are represented by the number of genes in the category, selected on the basis of a *p*-value < 0.05.

**Figure 6 ijms-21-02562-f006:**
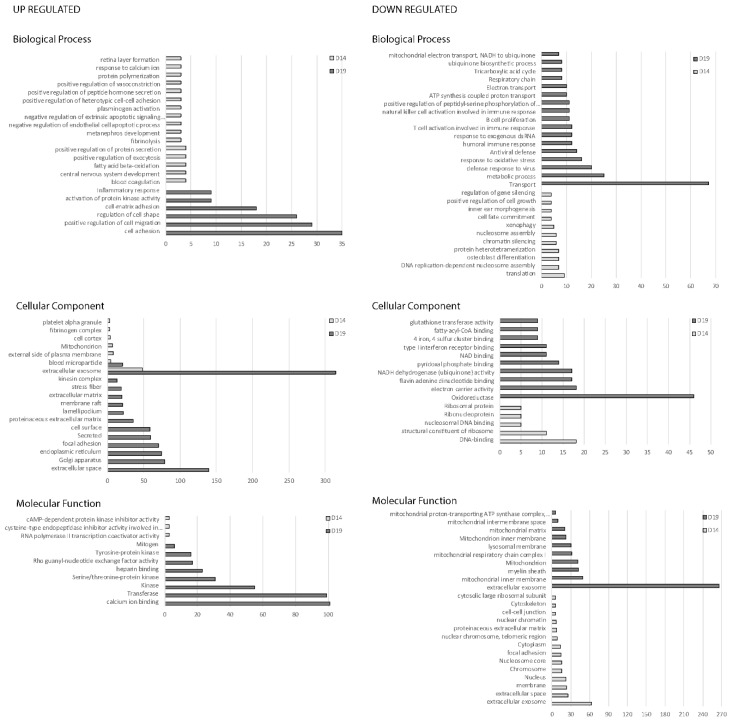
Gene ontology analysis for genes upregulated and downregulated in equine conceptus membrane, on day 14 and 19 of pregnancy between synchronous (recipient and donor ovulated on same day) and asynchronous (recipient ovulated 5 days after the donor) ET (D14 Syn/Asyn and D19 Syn/Asyn). The major categories within the biological process, cellular component and molecular function are represented by the number of genes involved in the category, selected on the basis of a *p*-value < 0.05.

**Figure 7 ijms-21-02562-f007:**
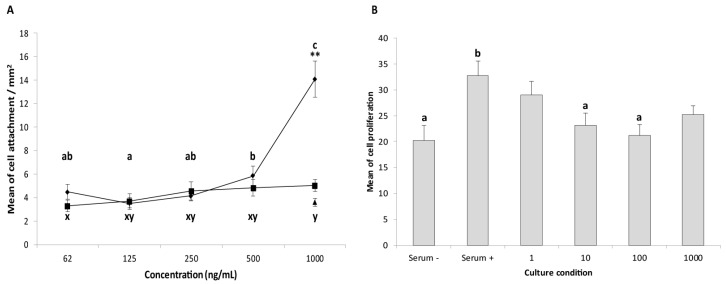
Effect of IGFBP3 on attachment (**A**) and proliferation (**B**) of cultured equine trophectoderm cells. For the attachment assay, wells were coated with BSA (negative control: 1000 ng/mL), fibronectin (positive control) or IGFBP3 (62, 125, 250, 500 and 1000 ng/mL). After 2 h of culture, the number of cells attached was counted and the average number of cells attached per mm^2^ (± s.e.m) calculated. The same letter indicates no significant difference (*p* > 0.05) in cell attachment in the presence of fibronectin (a,b,c) or IGFBP3 (x,y). Both fibronectin and IGFBP3 stimulated a dose-dependent increase in cell attachment, although the effect of the highest concentration of fibronectin was much larger. For the proliferation assay, cells were cultured for 30 h and then treated with IGFBP3 (1, 10, 100 and 1000 ng/mL), normal culture media (positive control), and serum-free media (negative control) for 24 h. BrdU was used to identify proliferating cells. The total number of cells (DAPI), and proliferating cells (BrdU positive) were counted, and the percentage of proliferating cells (± s.e.m) per mm^2^ was calculated. The analysis was performed for three independent experiments, with three replicates per condition; 5 non-overlapping sections were imaged per condition. Cell proliferation was higher in the presence of serum, but was not further increased by the addition of IGFBP3.

**Figure 8 ijms-21-02562-f008:**
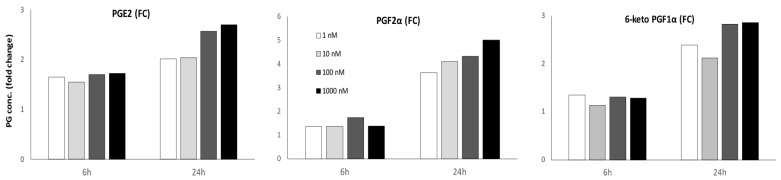
Effect of bradykinin on prostaglandin synthesis by equine trophectoderm cells. Cells were incubated in the presence of bradykinin (1, 10, 100 and 1000 ng/mL) for 6 and 24 h or with serum free medium for 6 h (negative control). The concentrations of PGE2, PGF2α and 6-keto PGF1α (pg/mL) were measured in the culture media, and are depicted as the fold change for a given treatment versus the negative control. The analysis was performed on three independent experiments, with three replicates per condition. Bradykinin stimulation resulted in increased production of all prostaglandins. After 24 h, there was evidence of a bradykinin dose dependent increase for PGE2 secretion (*p* < 0.0005) and a tendency for a dose dependent increase in PGF2α secretion (*p* = 0.055), but not in 6-keto PGF1α secretion.

**Figure 9 ijms-21-02562-f009:**
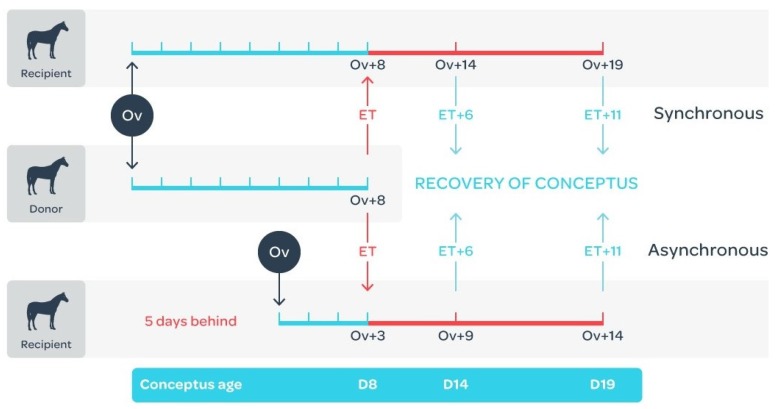
Day 8 embryos were transferred to recipient mares that ovulated on the same day (synchronous: *n* = 10) or 5 days after (asynchronous: *n* = 10) the donor mare. Conceptuses were recovered 6 or 11 days later (5 per group), when the conceptus would be at either 14 or 19 days of development and the recipient mares would be at either 9, 14 or 19 days after ovulation. Adapted from [40].

**Table 1 ijms-21-02562-t001:** The number of genes differentially expressed (DEGs) in the endometrium and conceptus membranes on day 14 and 19 of development following transfer of day 8 horse embryos to day 8 (synchronous) or day 3 (asynchronous) recipient mares. Genes with higher expression in the synchronous compared to the asynchronous pregnancies are referred to as upregulated and those with a lower expression are referred to as downregulated.

Endometrium	DEGs	Upregulated	Downregulated
D14–Syn/Asyn	523	174	349
D19–Syn/Asyn	715	283	432
D14–Syn/D19–Asyn	14	7	7
**Conceptus**			
D14–Syn/Asyn	556	285	271
D19–Syn/Asyn	3473	1870	1603
D14–Syn/D19–Asyn	3646	1961	1685

**Table 2 ijms-21-02562-t002:** Validation of the RNA-seq results by qRT-PCR for selected genes in the equine endometrium 6 (day 14) and 11 (day 19) days after transfer of day 8 embryos to the uterus of mares on day 8 (synchronous) or day 3 (asynchronous) after ovulation. Day 14 synchronous vs. asynchronous and day 19 synchronous vs. asynchronous. Positive values represent upregulated genes, and negative values represent downregulated genes in synchronous endometrium. Solute carrier: SLC; fold change: FC.

Gene Name	Gene Symbol	D14 RNA-seq		D14 PCR		D19 RNA-seq		D19 PCR	
		Log2 FC	P-Adjusted	Log2 FC	*p*-Value	Log2 FC	P-Adjusted	Log2 FC	*p*-Value
Interleukin 6 signal transducer	*IL6ST*	0.33	0.572	0.77	0.249	0.65	<0.001	0.27	0.283
Insulin receptor	*INSR*	0.96	0.009	0.58	0.291	0.99	<0.001	0.69	0.065
Leukemia inhibitory factor (LIF)	*LIF*	1.35	0.022	1.37	0.082	2.57	<0.001	3.07	0.001
LIF receptor	*LIFR*	0.27	0.419	1.22	0.148	−0.20	0.639	−0.44	0.879
Oxytocin receptor	*OXTR*	1.51	0.007	0.74	0.226	0.63	0.071	0.04	0.711
Prostaglandin F receptor	*PTGFR*	−1.47	<0.001	−1.83	0.016	−0.47	0.161	−1.46	0.153
SLC family 1, member 1	*SLC1A1*	0.08	0.962	−0.02	0.919	−1.49	<0.001	−1.75	0.002
SLC family 2, member 1	*SLC2A1*	1.89	<0.001	3.16	0.001	0.11	0.889	−1.32	0.379
SLC family 38, member 2	*SLC38A2*	0.36	0.283	0.09	0.636	1.10	<0.001	0.75	0.061
SLC family 43, member 2	*SLC43A2*	−0.47	0.425	−1.22	0.050	−0.33	0.450	−0.98	0.173
SLC family 5, member 1	*SLC5A1*	−0.28	0.752	−0.36	0.492	−0.08	0.935	−0.90	0.369

**Table 3 ijms-21-02562-t003:** Validation of the RNA-seq results by qRT-PCR for selected genes in equine conceptus membranes 6 (day 14) and 11 (day 19) days after transfer of day 8 embryos to the uterus of mares on day 8 (synchronous) or day 3 (asynchronous) after ovulation. Day 14 synchronous vs. asynchronous and day 19 synchronous vs. asynchronous. Positive values represent upregulated genes, and negative values represent downregulated genes in synchronous pregnancies. Solute carrier: SLC; fold change: FC.

Gene Name	Gene Symbol	D14 RNA-seq		D14 PCR		D19 RNA-seq		D19 PCR	
		Log2 FC	P-Adjusted	Log2 FC	*p*-Value	Log2 FC	P-Adjusted	Log2 FC	*p*-Value
**DNA-methyltransferase 3B**	*DNMT3B*	−0.71	<0.001	−0.64	0.048	0.12	0.613	0.66	0.018
**H19, Imprinted maternally expressed transcript**	*H19*	0.03	0.968	0.11	0.572	0.40	0.083	0.62	0.058
**Insulin-like growth factor 1**	*IGF1*	0.43	0.302	2.20	0.007	1.48	0.027	1.67	0.082
**Necdin**	*NDN*	−0.17	0.729	−0.53	0.405	1.42	0.001	1.57	0.019
**Paternally expressed gene 10**	*PEG10*	0.32	0.441	0.45	0.522	2.70	<0.001	2.89	0.004
**SLC family 1, member 4**	*SLC1A4*	0.79	0.008	1.10	0.002	0.72	0.003	0.77	0.062
**SLC family 1, member 5**	*SLC1A5*	−1.65	<0.001	−2.52	0.004	2.15	<0.001	3.07	0.029
**SLC family 2, member 3**	*SLC2A3*	0.22	0.305	0.54	0.167	−0.16	0.200	−0.02	0.680
**SLC family 2, member 5**	*SLC2A5*	0.49	0.012	1.55	0.002	−1.20	<0.001	−1.42	0.001
**SLC family 7, member 5**	*SLC7A5*	−0.04	0.941	0.69	0.464	1.64	<0.001	2.52	0.007
**Small nuclear ribonucleoprotein-associated protein**	*SNRPN*	−0.04	0.914	−0.03	0.772	−0.05	0.866	−0.40	0.542

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
