# Peer review of "Asynchronous Embryo Transfer Followed by Comparative Transcriptomic Analysis of Conceptus Membranes and Endometrium Identifies Processes Important to the Establishment of Equine Pregnancy"

_ijms, 2020, doi:10.3390/ijms21072562_

Round 1

Reviewer 1 Report

Scope of the article is hot topic in recent time, understanding of horse pregnancy establishment has several questionable moments till now.
Aim of this study was to determine the effect of synchronous or asynchonous day 8 embryos during preimplantation due to gene expression pattern at embryonic and uterine levels.
Title of manuscript is adequate and clearly describes the scope of the research.
Abstract is summarized sufficiently the importance of this topic and review the methods and findings of the experiment, the final conclusion underlined the impacts of different transcriptomic patterns between the two ET protocols.
Introduction is well connected and coherent with the experiment. Several way of molecular and physiological effects on preimplantation processes/success were explained and discussed.
Materials and methods
This part of the MS consists of ten sub-sections. All described analytical methods/techniques were fulfilled and served a high merit of research. Statistical analysis is clear.
Results
All the findings were described and demonstrated clearly.
figure 8. there is no value of x-axis.
Discussion
This part of the MS is accurate. All obtained results were supported and compared to previous research findings.
References were used and cited correctly.
Questions
"Two samples, 3 and 8, exhibited a very different pattern of expression to the other members of their respective groups (i.e. were outliers) and were therefore not included in subsequent statistical analyses."
You mentioned that the same animals were used in other experiment, were their any differences of the ultrasound investigation results of these samples (embryonic vesicle diameter or other evaluated parameters) compared to the the finally used samples?
Did you measure the size of the corpora luteum and P4 levels during the trial?
What do you think external P4 addition could eliminate this altered gene expression pattern?

I recommend to publish the manuscript in present form after answering the above questions.

Author Response

Thank you for your analysis and comments.

Figure 8 - the x axis is the fold change in the concentration of PGs. This is explained in the legend, but was not included to the figure.  A text box to label the x axis has been added to figure 8 (after revision figure 9)

With regard to the two outliers - samples 3 & 8. I have been back through the original data. The embryo size at transfer and vesicle size at time of collection did not differ significantly to other group members. Coincidentally, both of the embryos used were recovered from the same mare - the recipients from which the endometrium was collected were however different. Both recipients had 2 CLs - although we did not measure CL size not progesterone concentrations. At least one recipient mare per group also had two CLs - so this was not unique.

I have added a piece of text to explain that there were no clinical parameters to explain why these samples were outliers: Lines 316-320. "Looking back through the clinical data, these two samples were derived from day 8 embryos that were recovered from the same donor mare, but were comparable in size to all other day 8 embryos and gave rise to pregnancy vesicles that did not differ in size to other group members. Both recipient mares had two corpora lutea; however, so did at least one other mare per group, i.e. there were no clinical data to explain why these samples did not cluster with the others."

The question about exogenous P4 is interesting. I think that multiple ovulation or giving additional P4 in the first 2-3 days may make a difference - because it would advance the rise in the P4 concentration (i.e. equivalent to ovulating slightly earlier). However, with the ET induced difference being 5 days - I don't think any amount of exogenous P4 would remove the differences in gene expression observed. Articles by Willman et al (2011) and Beyer et al (2019) support the idea that additional P4 during early pregnancy (before day 14) does not affect gene expression or function much - but reducing P4 can.

I have added a line to the text (introduction - lines 86-87 ) to explain this and have included the extra citations given above (and renumbered the references throughout).

Reviewer 2 Report

Gibson et al in the present study examined the effects of asynchronous embryo transfer on the transcriptomic profile on early stage of the equine pregnancy. Additionally, the authors evaluated effect of  bradykinin on prostaglandin synthesis by equine trophectoderm cells.

The results presented in the manuscript constitute a very detailed and valuable analysis of the molecular basis of processes occurring in the early stages of pregnancy. Experiment correctly designed and carried out with the right methods.

A disadvantage of the experiment is small experimental group (n=5). Furthermore, I have a few questions and comments that will complement the manuscript and increase its quality.

1. To make easier to understand and follow of the scheme of the experiment, the authors should consider creating a Gantt chart, as in previous work (PMID: 28864009);

2. What quality of RNA samples is acceptable by the authors (both spectrophotometric and  BioAnalyzer results);

3. How were the starters designed (provide software). For example, in the case of the IGF1 gene, there are several variants of the transcript, whether a sequence alignment has been made to recognize all splice variants in PCR?

4. What algorithm was used for relative quantification (ΔΔCT Method or the E-Method with serial dilutions of cDNA mix of all the samples as relative standards)?

5. How did the authors select genes for validation using RT-qPCR? There is no information why these and no other genes are in Tables 2 and 3.

6. With reference to the previous question: From verse 630 in the discussion section, as well as in the abstract, the authors describe the potential importance of KNG1, IGFBP3 and FAM118B in pregnancy, highlight the changes observed; however, there is no validation of these results using RT-qPCR. Why the authors in this case are based exclusively on RNA-seq results?

Author Response

Many thanks for the comments and questions.

In response:

  1. We have included a modified figure to illustrate the production or pregnancies and collection of material (new Figure 1)
  2. The spectrophotometer analysis was at 260/280 to check for purity - if values were close to two, extraction was repeated. RNA quality was examined using the bioanalyzer - RIN values for conceptus samples all exceeded 8. For endometrial samples, values were lower and we decided to accept a cut-off of 6.6 (range 6.6-8.1). The RIN cut-off values have been added to the text. Line 177: "(RIN values > 8.0 for conceptus membrane samples and > 6.6. for endometrial samples)"
  3. The Primers were designed using PerlPrimer 1.1. When multiple splice variants were present for a given gene of interest, the primers were designed using mRNA coding sequence conserved across the variants. This has been added to the text. Lines 204-206: "Primers were designed using PerlPrimer 1.1; where multiple splice variants existed for a target gene, the primers were designed for areas of the mRNA that were conserved across the variants."
  4. Quantification of gene expression after qPCR was using the standard curve method, serial dilution of a known amount of cDNA was used. The text has been amended to explain this. Lines 211-214:  "a melting curve was performed to verify product specificity, and a 10-fold serial dilution of the target gene PCR product was amplified simultaneously with the samples to establish a standard curve that was used to quantify sample mRNA expression."
  5. Selection of genes for validation was based on genes used in previous studies for which we had already validated primers. Among these a selection was made for genes that showed a significant fold change on day 14 and/or 19 - or that did not change, so that we could test if the changes identified by sequencing were broadly repeatable by qPCR. An explanation has been added to the text. Lines 199-202: "To validate the sequencing results, RT-qPCR was performed for a number of genes for which we already had validated primers. A selection was made per tissue (conceptus and endometrium) to include genes that were upregulated, down-regulated and not altered by asynchrony at one or both time points."
  6. KNG1, IGFBP3 and FAM118B were indeed identified as being DEGs in all endometrial comparisons. The direction of expression change for FAM118B different for different comparisons and we did not try to design primers for FAM118B or further examine potential function.We were particularly interested in IGFBP3 and KNG1 because they were consistently upregulated in synchronous endometrium - but were unable to design successful primers (despite many attempts). This has been added to the text: Lines 453-4: "We were not able to design usable primers to validate the changes of expression for IGFBP3 and KNG1."    

Round 2

Reviewer 2 Report

The authors referred to all questions asked. I recommend acceptance of the manuscript in its present form.